# Distinct Responses to IL4 in Macrophages Mediated by JNK

**DOI:** 10.3390/cells12081127

**Published:** 2023-04-11

**Authors:** Luís Arpa, Carlos Batlle, Peijin Jiang, Carme Caelles, Jorge Lloberas, Antonio Celada

**Affiliations:** 1Biology of Macrophages Group, Department of Cellular Biology, Physiology and Immunology, Universitat de Barcelona, 08007 Barcelona, Spain; luisarpa@hotmail.com (L.A.); carlos.batlle3@gmail.com (C.B.); jin.jiang017@gmail.com (P.J.); 2Institute of Biomedicine, Universitat de Barcelona (IBUB), 08028 Barcelona, Spain; ccaelles@ub.edu; 3Department of Biochemistry and Physiology, School of Pharmacy and Food Sciences, Universitat de Barcelona, 08028 Barcelona, Spain

**Keywords:** monocytes/macrophages, chemokines, cytokines, kinases/phosphatases, inflammation

## Abstract

IL(Interleukin)-4 is the main macrophage M2-type activator and induces an anti-inflammatory phenotype called alternative activation. The IL-4 signaling pathway involves the activation of STAT (Signal Transducer and Activator of Transcription)-6 and members of the MAPK (Mitogen-activated protein kinase) family. In primary-bone-marrow-derived macrophages, we observed a strong activation of JNK (Jun N-terminal kinase)-1 at early time points of IL-4 stimulation. Using selective inhibitors and a knockout model, we explored the contribution of JNK-1 activation to macrophages’ response to IL-4. Our findings indicate that JNK-1 regulates the IL-4-mediated expression of genes typically involved in alternative activation, such as *Arginase 1* or *Mannose receptor*, but not others, such as *SOCS* (suppressor of cytokine signaling) *1* or *p21^Waf−1^* (cyclin dependent kinase inhibitor 1A). Interestingly, we have observed that after macrophages are stimulated with IL-4, JNK-1 has the capacity to phosphorylate STAT-6 on serine but not on tyrosine. Chromatin immunoprecipitation assays revealed that functional JNK-1 is required for the recruitment of co-activators such as CBP (CREB-binding protein)/p300 on the promoter of *Arginase 1* but not on *p21^Waf−1^*. Taken together, these data demonstrate the critical role of STAT-6 serine phosphorylation by JNK-1 in distinct macrophage responses to IL-4.

## 1. Introduction

Interleukin-4 (IL-4) is a cytokine with functional pleiotropy that plays an important role in host defense in cells involved in innate (macrophages) and acquired immunity (T and B lymphocytes) [1].

Macrophages play a critical role in the resolution of inflammation. During the initial inflammatory reaction, macrophages, under the effects of Th (T helper) 1-type cytokines such as IFN (Interferon)-γ, become pro-inflammatory and secrete a large number of harmful molecules (e.g., NO (Nitric oxide), reactive oxygen species (ROS), enzymes, and immunomodulatory cytokines such as TNF (Tumor necrosis factor)-α). This process has been named classical activation or the acquisition of an M1 phenotype [2]. During the later stages of inflammation, macrophages become anti-inflammatory and constructive [3]. They are activated by Th2-type cytokines, such as IL-4, and through the degradation of arginine they produce proline and polyamines, the latter of which serve to rebuild the extracellular matrix. This process is known as alternative activation or the acquisition of an M2 phenotype [4].

Upon ligand binding, IL-4 signals through a receptor comprising either the IL-4 receptor α (IL-4Rα) and CD132 (γc) chains (type I receptor) or IL-4Rα and IL-13Rα1 chains (type II receptor) [1]. Both types of chains oligomerize and, subsequently, JAK (Janus kinase) -1 and -3 are activated, inducing the phosphorylation of the IL-4 receptor. This process provides a docking site for STAT-6, which induces the phosphorylation of tyrosine 641 (Y641). This leads to its dimerization, translocation to the nucleus, and binding to specific response elements on target genes [1]. In addition, previous data have revealed that multiple serine residues are susceptible to phosphorylation on the STAT-6 transactivation domain (TAD) [5,6].

Several publications have shown that MAPK family members were activated by IL-4. Depending on the cell type, ERK (Extracellular signal-regulated kinase) in T cells [7,8], p38 in B cells [9], and JNK in fibroblasts [10] have been involved in signal transduction to this cytokine.

MAPKs are conserved serine/threonine kinases involved in the transduction of signaling that regulate cell growth, differentiation, and apoptosis [11,12,13]. These kinases include ERK-1 and -2, JNK-1 and -2, and p38. Through phosphorylation, MAPKs directly regulate downstream targets, including protein kinases, cytoskeleton components, phospholipase A2, and transcription factors or complexes, such as Ets (E-twenty-six)-1, Elk/TCF, and AP-1 (activating protein-1). In turn, these transcription factors promote immediate early gene expression.

In this study, we observed an early and strong activation of JNK-1 during macrophage response to IL-4 as well as the weak and late activation of ERKs and p38. The inhibition of JNK-1 activation resulted in the decreased expression of a number of genes typically induced by IL-4, such as *Arginase 1* or *Mannose receptor*, but not others, such as *SOCS1* or *p21^Waf−1^*. We have observed that STAT6 was phosphorylated at Y641 and serine. Tyrosine phosphorylation is independent of JNK-1, while serine phosphorylation is dependent on the aforementioned kinase. By using chromatin immunoprecipitation assays, STAT-6 and JNK-1 were detected in some promoters but not in others. This finding demonstrates the critical role of the serine phosphorylation of STAT-6 by JNK-1 in macrophages’ response to IL-4.

## 2. Materials and Methods

### 2.1. Reagents

Recombinant IL-4 and M-CSF were purchased from R&D Systems. SP600125, PD98059, and SB203580 were obtained from Calbiochem. Actinomycin D (Act D) and 5,6-dichlorobenzimidazole 1-β-D-ribofuranoside (DBR) were obtained from Sigma-Aldrich. The following antibodies used were used: anti-ERK-1/2, anti-phospho-p38 (Thr180/Tyr182), anti-JNK1, anti-STAT-6, phosphorylated anti-STAT-6 (Tyr 641), anti-phosphoserine, anti-CBP/p300, and anti-β-actin (Appendix A).

### 2.2. Cell Culture and Animal Models

Bone-marrow-derived macrophages (BMDMs) were obtained from 8-week-old C57BL/6 female mice (Charles River Laboratories, Wilmington, MA, USA), as described previously [14]. Bone marrow cells were extracted from femora, tibia, and humerus. The obtained cells were grown on plastic tissue culture dishes (150 mm) in DMEM (Cultek, Madrid, Spain) containing 20% FCS (GIBCO, Thermo Fisher Scientific, Waltham, MA, USA) and 20 ng/mL of recombinant M-CSF) (Thermo Fisher, Loughborough, England) supplemented with 100 U/mL of penicillin and 100 µg/mL of streptomycin. In a humidified atmosphere, cells were incubated at 37 °C with 5% CO_2_. After 7 days of culture, a homogeneous population (99.34 ± 0.52% CD11b/CD18 and 98.41 ± 0.93% F4/80) of adherent macrophages was obtained. BMDMs were left for 16 h in medium without M-CSF to allow for synchronization of cell cycles prior to stimulation. Mice deficient in JNK-1 (JNK-1^−/−^) [15] were donated by Dr. R. A. Flavell (Yale University School of Medicine, New Haven, CT, USA). The Animal Research Committee of the University of Barcelona approved use of animals (number 2523).

### 2.3. RNA Extraction, Reverse Transcription PCR, and qPCR

RNA extraction was achieved using a method previously described by our group [16]. To clone the reporter plasmids and perform PCR, total RNA from cells was purified using the ReliaPrep RNA Miniprep System (Promega, Madison, WI, USA). To remove contaminating DNA, RNA was treated with DNase (Roche, Basel, Switzerland). Using the Moloney murine leukemia virus (MMLV) reverse transcriptase, RNase H Minus (Promega, Madison, WI, USA), RNA was retrotranscribed into cDNA according to the manufacturer’s indications. Quantitative PCR (qPCR) was performed using the SYBR Green Master Mix (Applied Biosystems, Waltham, MA, USA). To design the primers, we used Primer3Plus (https://www.bioinformatics.nl/cgi-bin/primer3plus/primer3plus.cgi (accessed on 12 June 2022). For each gene, water was used as negative control. When a signal was detected in these negative controls (at 40 Ct), the primer pairs were replaced with alternative ones. By making a standard curve from serially diluted cDNA samples, we calculated the amplification efficiency for each pair of primers. We only used primer pairs with an amplification efficiency of 100 ± 10%. Appendix A provides a list of the primers used (Sigma Aldrich, St. Louis, MO, USA). The ΔΔCt method [17] was used to analyze the data. This was performed using the Biogazelle Qbase^+^ software. Gene expression of the three housekeeping genes, namely, *Hprt1*, *L14*, and *Sdha*, was used to normalize data to address the one-sample problem. The reference genes’ stability was determined by establishing that their geNorm M value was inferior by 0.5 [18].

### 2.4. Protein Extraction and Western Blot Analysis

Protein extraction was accomplished as described in our previous work [19]. In cold PBS, cells were washed twice and lysed on ice using lysis solution (1% Triton X-100, 10% glycerol, 50 mM HEPES at pH 7.5, 250 mM NaCl, 1 µg/mL aprotinin, 1 µg/mL of leupeptin, 1 µg/mL of iodoacetamide, 1 mM PMSF, and 1 mM sodium orthovanadate). Then, through centrifugation at 13,000× *g* for 8 min at 4 °C, we removed the insoluble material. In Laemli SDS-loading buffer, cell lysates (50–100 µg) were boiled at 95 °C. Subsequently, cell lysates were separated by 10% SDS-PAGE. Then, proteins were transferred electrophoretically to nitrocellulose membranes (Hybond-ECL, Amersham, England). Next, for 1 h at room temperature, membranes were blocked in 5% dry milk in TBS-0.1% Tween 20 (TBS-T). When using the anti-phosphoserine antibody, we did not employ milk as a blocking agent because milk casein is phosphorylated at several serine residues. Instead, we used bovine serum albumin as recommended by the supplier (Abcam, Cambridge, UK). Membranes were incubated with primary antibody overnight at 4 °C (Appendix A). Subsequently, membranes were washed three times in TBS-T. This was followed by incubation with horseradish-peroxidase (HRP)-conjugated secondary antibody for 1 h at room temperature. After three 5 min washes with TBS-T, chemiluminescence detection was performed (Amersham), and the membranes were exposed to X-ray films (Amersham).

### 2.5. JNK Activity Assay

JNK activity was measured as previously described [20]. Nuclear extracts were obtained from cells and then immune-precipitated with protein A-sepharose and anti-JNK-1 antibody. After five washes, the reaction was performed with 1 µg of cytosolic glutathione S-transferases (GST)-c-jun (1-169) (MBL) as JNK substrate, 20 µM ATP and 1 µCi µ^32^P-ATP. In Figure 6, protein A-sepharose and anti-STAT-6 antibodies immune-precipitate total protein extracts (150 µg) from macrophages. Subsequently, immune-precipitates were washed, and used as substrate for JNK-1 instead of GST-c-jun. Then, SDS-PAGE electrophoresis was performed, and the gel was exposed to Agfa X-ray films.

### 2.6. Chromatin Immunoprecipitation Assay

The ChIP assays were performed as described in our previous work [21,22]. BMDMs were incubated with the recommended stimuli and time. Subsequently, 20 × 10^6^ cells were cultured in a 150 mm plate and fixed in paraformaldehyde. After 10 min at room temperature, to stop fixation, glycine (2 M) was added. After 5 min, the plates were washed, and the cells were then scraped and recovered. The precipitate was washed in 1 mL of PBS, Buffer I (10 mM HEPES at pH 6.5, 0.25% Triton X-100, 10 mM EDTA, and 0.5 mM EGTA), and Buffer II (10 mM HEPES at pH 6.5, 20 mM NaCl, 1 mM EDTA, and 0.5 mM EGTA). A protease inhibitor cocktail (1 mM PMSF, 1 mM iodoacetamide, 1 mM sodium orthovanadate, 10 µg/mL of aprotinin, and 1 µg/mL of leupeptin) was added before centrifugation to Buffer I and Buffer II. A total of 300 µL of lysis buffer (1% SDS, 10 mM EDTA, 0.5 mM Tris-HCl at pH 8.1, and the protease inhibitor cocktail) was added to the pellet of cells and incubated at RT. Then, the samples were sonicated for 10 min in high mode (30″ on/30″ off) using Bioruptor Twin (Diagenode; Liege, Belgium). The procedure was repeated 5 times. Subsequently, to confirm a good degree of sonication of the samples (the DNA fragments should have a size of 200 bp to 1200 bp), DNA agarose gel electrophoresis was performed. The soluble chromatin was centrifuged at 16,000× *g* for 10 min and diluted to a final volume of 1.1 mL in the following buffer (1% Triton X-100, 2 mM EDTA, 150 mM NaCl, and 20 mM Tris-HCl at pH 8.1, and a protease inhibitor cocktail). For control or INPUT, 100 µL was separated and stored at 4 °C. To reduce the number of non-specific bindings, the remaining sample was incubated overnight at 4 °C with 2 µg of sonicated salmon sperm DNA (Amersham), 2.6 µg of non-specific IgGs (Sigma Aldrich, St. Louis, MO, USA), and 20 µg of Magna ChIP protein A magnetic beads (Millipore, Burlington, MA, USA). To remove the beads, the sample was centrifuged at 16,000× *g* for 10 s. Then, the sample was diluted to a volume of 2 mL (1 mL of the specific precipitate and 1 mL of the control). The two precipitates were incubated for 6 h with the same amount of either antibody (phosphorylated anti-STAT-6 (Tyr 641) or anti-CBP/p300) or a non-specific IgG. Then, the samples were incubated at 4 °C overnight with 20 µL of magnetic beads. The following day, the samples were centrifuged (16,000× *g* for 10 s), and the beads were washed and incubated for 10 min in 1 mL of TSE I (150 mM NaCl, 0.1% SDS, 1% Triton X-100, 2 mM EDTA, and 20 mM Tris HCl at pH 8.1); in 1 mL of TSE II (500 mM NaCl, 0.1% SDS, 1% Triton X-100, 2 mM EDTA, and 20 mM Tris HCl at pH 8.1); and finally in 1 mL of Buffer III (0.25 M LiCl, 1% NP-40, 1% *w*/*v* deoxycholate, 1 mM EDTA, and 10 mM Tris HCl at pH 8.1). After these washes, the beads were cleaned with 1 mL of PBS (4 °C), and the immune precipitates were eluted with 300 µL of the following solution (0.1 M NaHCO3 and 1% SDS). The elution was performed in three steps. First, the beads were incubated for 20 min in 100 µL of the elution solution. Subsequently, the samples were centrifuged at 16,000× *g* for 10 s. The resulting supernatant was recovered in a 1.5 mL Eppendorf tube. This procedure was repeated two more times and a final volume of 300 µL was obtained. Before DNA purification, a “reverse crosslinking” step was required, wherein the samples (non-specific and immune precipitates) and INPUTs were incubated overnight at 65 °C. The following day, the QIAquick PCR Purification Kit (Qiagen, Hilden, Germany) was used to purify the DNA of the samples. The final elution volume was 30 µL. These samples were analyzed by qPCR using the primers of the *Arginase 1* promoter: forward, 5′-GCATTGTTCAGACTTCCTTATGCTT-3′; reverse, 5′-TGTTGGCTAATACAGCCTG-TTCAT-3′ [23]. For the control, we used a non-promoter region of an unrelated gene, the *36B4* gene encoding a ribosomal protein. The following primers were used: 5′-AGATGCAGCAGATCCGCAT-3′ and 5′-GTTCTTGCCCATCAGCACC-3′. Primers used for PCR amplification of the *p21^Waf−1^* promoter were 5′-TTAACGCGCGCCGGTTCTA-3′ and 5′-AGCGCATTGCTACGGGGAA-3′ [24,25].

To obtain the final results, we performed two normalization steps. The first step involved the specific INPUTs and the second one involved the results obtained from the analysis of the *36B4* gene encoding a ribosomal protein, which was located outside the promoter region of *Arginase 1* or *p21^Waf−1^*.

### 2.7. Statistical Analysis

Data were analyzed using the Student’s *t*-test. Statistical analysis was performed with the GraphPad Prism 9.1 software.

## 3. Results

### 3.1. IL-4 Induces Early and Short Activation of JNK-1 but Not of ERK or p38

A number of publications have shown that depending on the cell type, ERK in T cells [7], p38 in B cells [26], and JNK in fibroblasts [27] are involved in signal transduction to IL-4. Based on these findings, we addressed whether MAPK activation is involved in the IL-4-mediated alternative activation of bone-marrow-derived macrophages. For this purpose, primary macrophages obtained from murine bone marrow were deprived of their specific growth factor (M-CSF) for 18 h to minimize MAPK activity; then, they were stimulated with IL-4 for the indicated periods of time (Figure 1). The activity of JNK-1, reported as glutathione S-transferase (GST)-c-jun, was strongly induced after 5 min of IL-4 treatment and was maintained for only 15 min (Figure 1A), thereby suggesting that JNK-1 participates in the alternative activation of macrophages. The activity of JNK-2 was also measured but was undetectable in in vitro kinase assays (data not shown). In contrast to JNK-1, the Western blot analysis of both phospho-ERK-1/2 and phospho-p38 revealed no activation at early stages but a significant induction of both kinases after 60 min of IL-4 stimulation (Figure 1B,C).

To determine whether there is a negative feedback mechanism induced by IL-4 to regulate MAPK activity, we analyzed the expression of MAPK phosphatases (MKP) 1, 2, and 5 as well as PAC1 and CPG 21. In contrast to M-CSF, which activates MAPKs and induces the expression of several members of the MKP family, IL-4 was only able to induce the expression of MKP-2 and, very transiently, MKP-5 (Figure 2). These results show a correlation between the dephosphorylation state of JNK and the induction of some specific MKPs [28,29].

### 3.2. IL-4-Induced JNK-1 Activation Contributes to the Regulation of Selective Genes

Next, we evaluated the involvement of JNK-1 in the alternative activation of macrophages mediated by IL-4. For this objective, we analyzed the expression levels of several genes, including *Arginase 1*, chemokines such as *CCL22* (a macrophage-derived chemokine) and *CCL24* (eotaxin-2), the cytokine *IL-10*, the *Mannose Receptor*, the *scavenger receptor CD163*, the *suppressor of cytokine signaling (SOCS)-1*, and the regulators of the cell cycle *p21^Waf−1^* and *c-myc*. In previous studies [30], we determined the time course of the induction of these genes by IL-4. Most were induced at high levels within 3 h after IL-4 treatment and maximal induction was detected after 6 h. The expression of *c-myc* and *SOCS1*, in contrast to the other genes, was detected early, namely, within the first 1 to 3 h after treatment.

To determine the role of JNK-1 in IL-4-induced gene expression, we used the selective inhibitor SP600125 and the JNK-1 knockout mouse model. Previous studies conducted by our group demonstrated that the dose of SP600125 used in the macrophages in this study blocks JNK activity without inducing cellular toxicity [31]. Surprisingly, the inhibition of JNK with SP600125 resulted in the efficient blockage of the expression of a subset of genes, including *Arginase 1*, *Mannose Receptor*, *CD163*, and *c-myc*; the chemokines *CCL22* and *CCL24*; and the cytokine *IL-10* (Figure 3A), whereas the expression of *SOCS1* or *p21^Waf−1^* was not significantly reduced (Figure 3B), thereby suggesting that the link between JNK-1 and IL-4 responses may be promoter-dependent. We also performed similar experiments using SB203580 to inhibit p38 and PD98059 to block MEK and, therefore, ERK-1/2 activity; however, none of these inhibitors significantly reduced the expression of the genes tested (Appendix A). To confirm the role of JNK-1, we also used macrophages from JNK1^−/−^ mice. In these cells, the expression of *Arginase 1*, *CCL22*, *CCL24*, and *c-myc* was drastically downregulated (Figure 4A). However, the levels of *SOCS1* or *p21^Waf−1^* were not affected (Figure 4B).

### 3.3. JNK-1 Does Not Affect mRNA Stability

Previous studies reported that MAPKs perform posttranscriptional regulation by affecting the stability of specific mRNAs [31,32]. Therefore, we tested whether the effects observed on the expression levels of the genes studied herein were due to the JNK-dependent modulation of their mRNA stability. We first induced the expression of IL-4-regulated genes and then blocked further mRNA synthesis by using a cocktail of Actinomycin D and 5,6-dichlorobenzimidazole 1-β-D-ribofuranoside (DBR) [33], at a concentration sufficient to block all further RNA synthesis, as determined by [3^H^]UTP incorporation [34]. We then measured the remaining mRNA for each gene after different periods of time. To normalize the results of each time point, for each treatment, we set the level of expression at 100% in the absence of an inhibitor. We did not detect any significant variation in the mRNA stability of the genes when the cells were pretreated for 1 h with the JNK inhibitor SP600125 before the addition of IL-4 (Figure 5). This observation suggests that JNK-1 affects the expression of these genes at the transcriptional level rather than their mRNA stability.

### 3.4. JNK-1 Phosphorylates STAT-6 on Serine Residues without Affecting Its Binding to DNA

STAT-6 must be phosphorylated on Y641 to induce its dimerization, translocation to the nucleus, and binding to target genes [1]. To determine the degree of phosphorylation on Y641, we checked whether the activity of JNK-1 toward STAT-6 interfered with the JAK-mediated tyrosine phosphorylation of STAT-6. For this purpose, we stimulated cells with IL-4 for 15 min in the presence or absence of the JNK-1 inhibitor SP600125. Having stimulated the cells, we immunoprecipitated STAT-6 and performed an immune-blotting assay against STAT6 phosphorylated on Y641. No variations were observed in the phosphorylation of STAT-6 on tyrosines (Figure 6A). The DNA-binding capacity of STAT-6 was tested through chromatin immunoprecipitation assays using the promoter of *Arginase 1*. As described previously [35,36], STAT-6 bonded to the *Arginase 1* promoter (Figure 6B). No impaired binding of STAT-6 was observed when JNK-1 was inhibited in the IL-4-stimulated cells (Figure 6B). These data demonstrate that Y641 phosphorylation is not mediated by JNK; therefore, this kinase could be involved in another phosphorylation process.

**Figure 6 cells-12-01127-f006:**
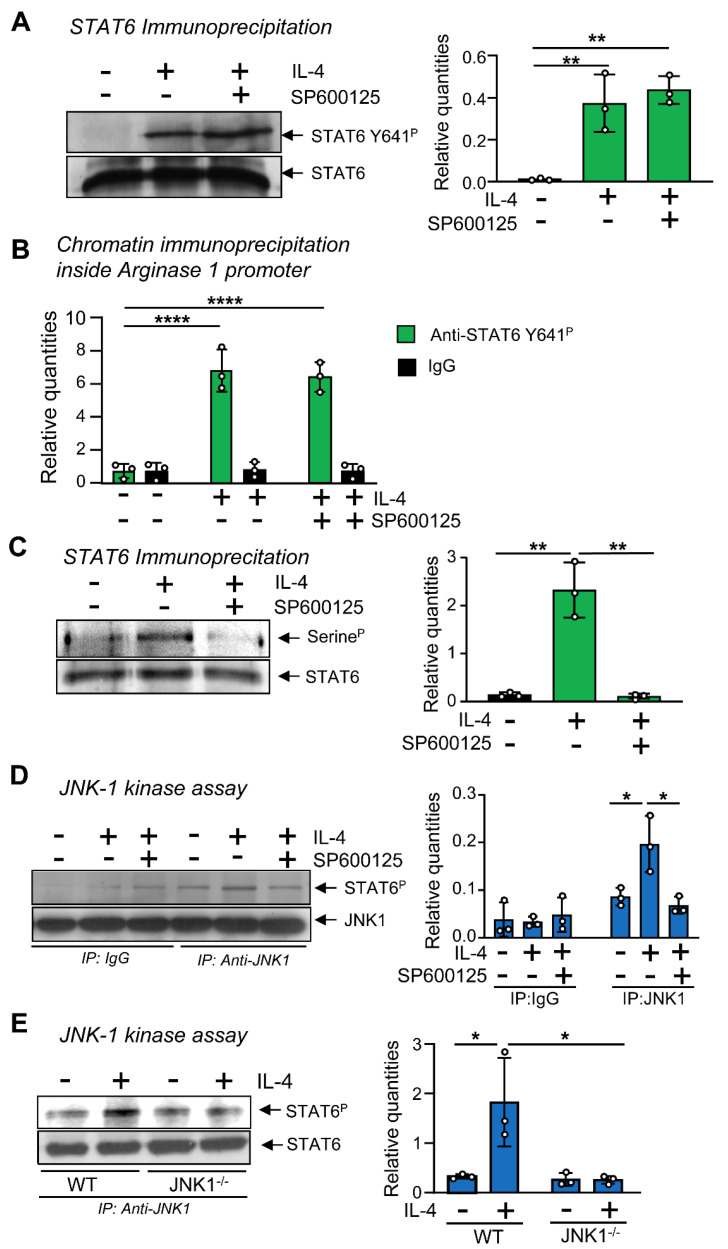
JNK-1 phosphorylates STAT-6 on serine without affecting its capacity to bind DNA. For A to C, macrophages were pretreated with the JNK inhibitor SP600125 (SP) for 1 h and then stimulated with IL-4 for 15 min. (**A**) Phosphorylation of STAT-6 (Y641) was analyzed by immunoprecipitation of STAT-6 and then via immunoblotting with an antibody, namely, either anti-phospho-Stat6 (Y641) or anti-STAT-6. (**B**) Chromatin immunoprecipitation assay (CHIP) was performed using the antibodies indicated. The presence of STAT6 Y641P in the *Arginase 1* promoter was evaluated using qPCR and normalized with the level of expression of a *36B4* exon and the inputs of each sample as a loading control. (**C**) Phosphorylation of STAT-6 on serine was analyzed by immunoprecipitating STAT-6 and then via immunoblotting with an antibody against phospho-serine or anti-STAT6. (**D**) Quiescent macrophages were stimulated with IL-4 for 15 min to reach maximum JNK-1 activity and then total protein extraction was performed. STAT-6 from quiescent macrophages (to avoid any basal kinase activity on STAT) was immunoprecipitated (150 μg of total protein extracts) and used as substrate in an in vitro kinase assay for JNK-1. As control for immunoprecipitation, IgG was used. An immunoblot for JNK1 was performed in parallel as a load control for the kinase assay. (**E**) An experiment similar to (**D**) but in which macrophages derived from WT or JNK-1 deficient mice (JNK-1^−/−^) were used. As a control for charge, a sample of total protein extracts was used for immunoblotting with an antibody (anti-STAT6). The results are shown as the mean ± SD of 3 independent experiments. * *p* < 0.05, ** *p* < 0.01, and **** *p* < 0.0001 in relation to the corresponding treatments after all the independent experiments had been compared. Data were analyzed using Student’s *t*-test.

Regarding STAT6′s activation, it has recently been described that in addition to Y641, STAT6 requires the phosphorylation of S407, which is located in the DBD (DNA-binding domain) [37]. Therefore, we examined whether STAT-6 is a substrate for JNK-1. Due to the lack of commercial antibodies that can detect STAT-6 phosphorylated on specific serines, we tested whether JNK-1 could phosphorylate STAT-6 on serine. For this purpose, we immunoprecipitated STAT-6 from IL-4-stimulated cells in the presence or absence of SP600125 and immunoblotted it with an anti-phophoserine antibody. The STAT-6 from cells induced with IL-4 showed strong phosphorylation on serine (Figure 6C). Interestingly, in cells treated with both IL-4 and SP600215, we did not detect any serine phosphorylation on STAT-6. This observation suggests that JNK is responsible for this phosphorylation. Moreover, to confirm these data, we immunoprecipitated STAT-6 from quiescent macrophages and used it as substrate in a JNK-1 kinase assay (Figure 6D). Based on the time course of JNK activation (Figure 1A), JNK-1 was immunoprecipitated from cells stimulated with IL-4 for 15 min. In the cells treated with IL-4, phosphorylated STAT-6 co-immunoprecipitated with JNK. Treatment with SP600125 reduced this effect (Figure 6D), which was more evident when we used the JNK-1^−/−^ cells (Figure 6E). So far, all these data suggest that although JNK-1 mediates the serine phosphorylation of STAT-6, it does not modify the phosphorylation of STAT-6 on tyrosine or its capacity to bind DNA.

### 3.5. JNK-1 Is Required for Promoting the Recruitment of CBP/p300 to the Arginase 1 Promoter

The phosphorylation of serine 727 of STAT-1 is responsible for the recruitment of cofactors at the promoter level [38]. IL-4 induces the phosphorylation of the IL-4α receptor, which recruits JAK and STAT6 for phosphorylation. Phosphorylated STAT6 triggers the formation of dimers and, subsequently, the translocation of dimerized STAT6 into the nucleus for transcriptional regulation after the recruitment of coactivators to the transcriptosome, such as CBP/p300 or the nuclear receptor coactivator 3 (NCOA3) [39,40]. Since the interaction between JNK-1 and STAT-6 resulted in the serine phosphorylation of STAT-6, we evaluated whether this interaction could also be a mechanism for cofactor recruitment. For this purpose, we performed chromatin immunoprecipitation assays.

First, we tested whether CBP/p300 binds to the *Arginase 1* promoter in our macrophage model, as described before in other types of cells [41,42]. We stimulated quiescent macrophages with IL-4 for 15 min. Using chromatin immunoprecipitation assays, we observed that the treatment with IL-4 induced the binding of CBP/p300 to the *Arginase 1* promoter, which was reversed by SP600125 (Figure 7A). To confirm these results, we used the JNK-1^−/−^ model. We stimulated the cells with IL-4 for 15 min and performed chromatin immunoprecipitation assays. In JNK-1^−/−^ cells, after stimulation with IL-4, no increase in the binding of CBP/p300 to the Arginase 1 promoter was observed (Figure 7B). Moreover, we also examined the recruitment of CBP/p300 in the promoter of *p21^Waf−1^*, whose expression is not inhibited in the absence of JNK-1 (Figure 3). In this case, we still detected the binding of CBP/p300 in the JNK-1^−/−^ cells treated with IL-4 (Figure 7B). These data suggest that JNK-1 activity is required for the recruitment of cofactors in some IL-4-induced genes.

## 4. Discussion

The involvement of MAPKs in the cell-type-dependent-signaling of IL-4 by ERK in T cells [7], p38 in B cells [26], and JNK in fibroblasts [27] has been previously documented. In our study, we determined the activation of MAPKs in macrophages activated by IL-4 and how JNK regulates the macrophage response to this cytokine. We did not explore the upstream regulators of JNK-1 in response to IL-4. However, we found that STAT-6, a critical mediator of IL-4 signaling, is phosphorylated at tyrosine 641, which occurs through the action of the kinase JNK-1 in serine. Due to a lack of commercially available reagents, we were unable to determine the exact serine phosphorylation site on STAT-6. However, although the activation of JNK-1 is required for the maximal expression of several genes, it is not necessary for STAT-6 translocation to the nucleus and DNA binding. The cross-talk between STAT-6 and JNK-1 provides a mechanistic link through which cytokine signaling can be modulated.

In our studies, the involvement of JNK-1 appears to play a critical role in the regulation of the expression of several genes induced by IL-4, as demonstrated by the relatively broad effects of the JNK-1 inhibitor SP600125. The effect of some MAPKs, such as p38 in IFN-γ-inducible genes, has been associated with the regulation of mRNA stability [31,43]. Using synthetic blockers of RNA synthesis, we have demonstrated that this is not the case for the effects of JNK-1 on the IL-4-dependent genes studied herein. Therefore, these observations suggest that during the macrophage response to IL-4, JNK-1 serves to modulate transcriptional events and enhance the expression of selective targets. Studies in STAT-6-deficient mice [44,45] showed that STAT-6 is involved in a highly confined manner in the signaling carried out by IL-4, playing a critical role in generating many of the responses induced by IL-4. However, whereas IL-4-induced differentiation appears to be largely dependent on STAT-6, IL-4-induced proliferation and survival have been shown to be at least partially independent of STAT-6 [44,45]. This finding suggests that IL-4 uses additional pathways other than STAT-6 to regulate gene expression. This does not seem to be the case here, as the genes whose regulation is affected by JNK-1 depend only on STAT-6 activation [30].

Our results confirm and extend the previous observations of Haoa et al. [46], showing the involvement of JNK signaling in IL-4. However, these authors used the two cell lines RAW264.7 and THP-1 as a cellular model of macrophages, while we used primary cultures of macrophages. In addition, we showed the critical role of the serine phosphorylation of STAT6 in the transactivation of several genes. Finally, we confirmed our previous observations showing that gene induction by IL-4 does not have a common signaling mechanism. Thus, as we reported previously, the deacetylation of C/EBPβ inhibited the IL-4-induced expression of *Arginase-1*, *Fizz1*, and *Mannose receptor*, while in other genes, such as *Ym1*, *Mgl1*, and *Mgl2*, expression was not affected [47].

One question that remains to be resolved is the location of the phosphorylated serine in STAT-6. In the literature, the IL-4-induced serine phosphorylation of STAT-6 is a highly controversial topic whose conclusions greatly depend on the experimental conditions and, in particular, the cell type used. Using Ramos cells (a B cell originating from Burkitt’s lymphoma), Pesu et al. [48] showed that IL-4-induced transcription requires the serine phosphorylation of STAT-6. In HeLa cells (a human cell derived from adenocarcinoma), Shirakawa et al. [49] demonstrated that the cytokine IL-1 mediated by JNK induces STAT-6 phosphorylation at serine 707. This phosphorylation decreases the DNA-binding ability of IL-4-stimulated STAT6, which has been reported to be a mechanism controlling the balance between IL-1 and IL-4 signals.

Recently, using multiple human cell lines of fibroblasts, the activation of STAT6 has been shown to be critical in antiviral innate immunity [37]. In this case, the phosphorylation of serine 407 located in the DNA-binding region plays a determining role. However, studies of the structural basis for DNA recognition by STAT6 show that the residue S407 is not likely to be accessible for phosphorylation by any kinase in the conformations of the protein observed in the crystal structures where STAT-6 is bound to DNA [50]. This controversy intensified when the same authors used luciferase-reporter-based assays to show that the S407 mutation nullifies the IL-4 response [50]. In fact, a large number of proteins, including CBP/P300, CD28, C/EBPβ, Detergent-sensitive factor, Ets-1, glucocorticoid receptor (GR), IFNαRI, IL-4Rα, IRF4, LITAF, NF-kB, p100, PU.1, SRC-1, and STAT-2, have been reported to interact with STAT-6 [51], and the binding of some of these proteins may undergo conformational changes to STAT-6 that render S407 capable of being phosphorylated.

The mechanism of JNK-1-enhanced gene expression remains elusive. However, we have demonstrated that the binding of CBP/p300 to the *Arginase 1* promoter was increased in the presence of JNK-1. It has been described that CBP/p300 must be serine-phosphorylated to act as a co-activator [52,53], and in some studies, this phosphorylation was carried out by members of the MAPK family [54]. It has been proposed that STAT-6 is acetylated by CBP/p300 [55]. The acetylation of STAT-6 was shown to be required for the STAT-6-mediated activation of expression [55,56].

The binding of STAT-6 to DNA alone is not normally sufficient to stimulate a specific locus. The initiation of transcription requires the interplay of STAT-6 with the basic transcription machinery, which is dependent on different groups of transcriptional co-regulatory proteins. STAT-6 interacts with co-factors through its transactivation domain [57]. Although a direct physical interaction between STAT-6 and CBP/p300 has been demonstrated in some studies, the binding relies on the adaptor protein p100 [58]. p100 is another co-activator protein that recruits histone acetyltransferase activity to STAT6 and enhances STAT-6-mediated transcriptional activation and gene expression [59]. CBP/p300 binds to the p300/CBP co-integrating protein (p/CIP), also known as the nuclear receptor co-activator-3 (NCoA-3), thereby recruiting it into the STAT-6 transcriptional activation complex [40]. p/CIP belongs to the family of p160/SRC co-activator proteins and was found to be a positive regulator of transcriptional activation by STAT-6. A member of the p160/SRC family, SRC-1 (NCoA-1), was found to be crucial for activation by STAT-6. Unlike p/CIP, SRC-1 interacts directly with STAT-6. Finally, a collaborator of STAT-6 (CoaSt6)-associated Poly(ADP-ribose) polymerase activity has been shown to modulate STAT-6-dependent gene transcription [60]. On the basis of our results, we have demonstrated that JNK-1, a signal transduction molecule, is required to initiate the activation of some genes by IL-4.

## 5. Conclusions

In macrophages, after the interaction of IL-4 with its receptor, the phosphorylation of the STAT-6 molecule is induced in tyrosine 641, leading to its dimerization and translocation to the nucleus. For the induction of certain genes, such as *Arginase 1* or the *Mannose receptor*, the activation of JNK-1 and the phosphorylation of STAT-6 in serines are also required. Similarly, JNK-1 activation is necessary to recruit co-activators such as CREB-binding protein (CBP)/p300 to the promoters of these genes. However, for other genes, such as *p21^waf1^* or *SOCS1*, STAT-6 does not require serine phosphorylation nor the recruitment of co-activators. In conclusion, the transcription machinery induced by IL-4 is not the same for all genes.

## Figures and Tables

**Figure 1 cells-12-01127-f001:**
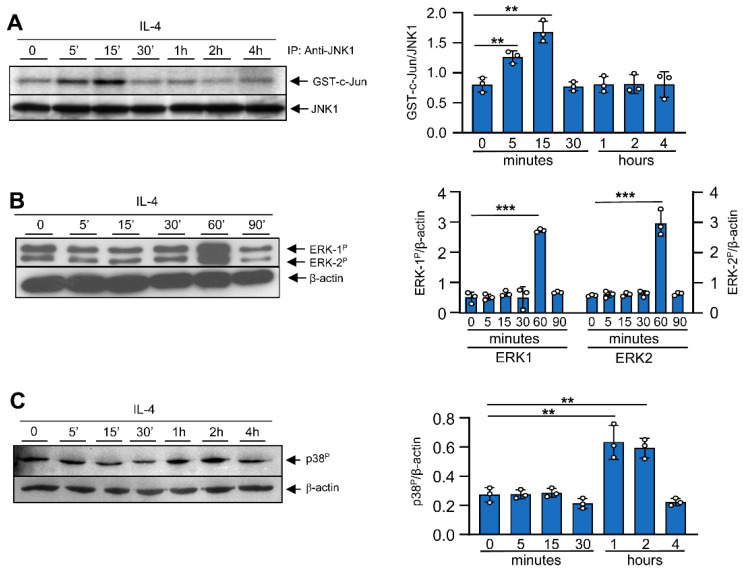
Effects of IL-4 on MAPK activation. Bone-marrow-derived macrophages were cultured for 6 days in the presence of M-CSF. Then, to render the cells quiescent, they were deprived of M-CSF for 18 h. At this point, IL-4 (10 ng/mL) or M-CSF (10 ng/mL) was added for the indicated periods of time. (**A**) JNK-1 activity was studied after immunoprecipitation and then an in vitro kinase assay was performed on recombinant c-Jun. An immunoblot for JNK-1 was performed in parallel as a loading control for the kinase assay. (**B**,**C**) Activation of MAPK ERK-1/2 and the phosphorylated form of p38 were analyzed via Western blot using the corresponding antibodies. In parallel, as a loading control, an immunoblot for β-actin was performed. Images on the right depict quantification by densitometry of 3 independent experiments. The results are shown as the mean ± SD. ** *p* < 0.01 and *** *p* < 0.001 in relation to the corresponding treatments with IL-4 after all the independent experiments had been compared. Data were analyzed using Student’s *t*-test.

**Figure 2 cells-12-01127-f002:**
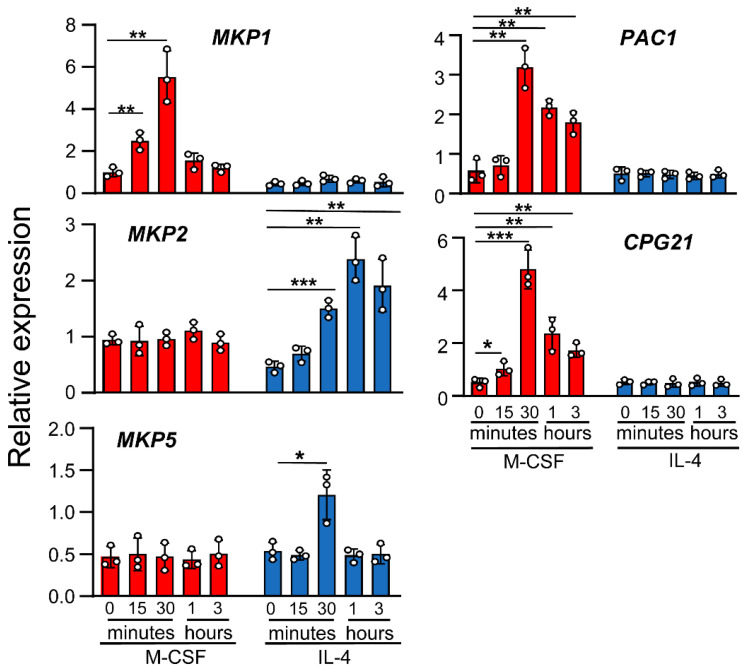
Effects of IL-4 on MKP expression. Macrophages were treated with M-CSF (control) or IL-4 for the indicated periods of time. MKP expression was analyzed by qPCR. The results are shown as the mean ± SD of 3 independent experiments. * *p* < 0.05, ** *p* < 0.01, and *** *p* < 0.001 in relation to the corresponding treatments after all the independent experiments had been compared. Data were analyzed using Student’s *t*-test.

**Figure 3 cells-12-01127-f003:**
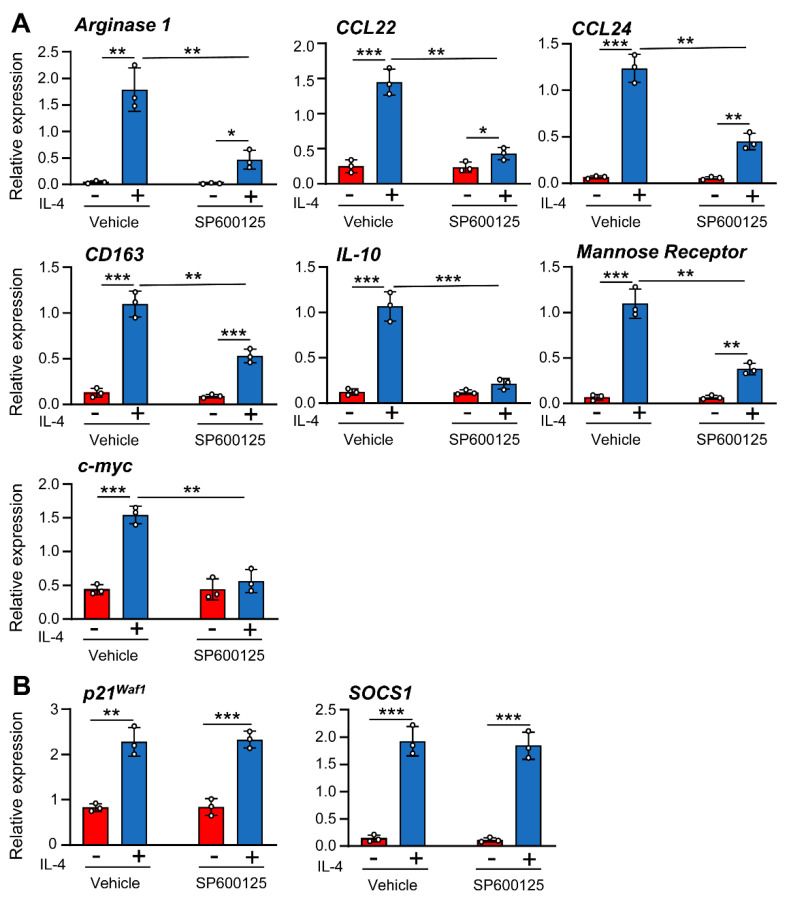
(**A**,**B**) Different effects of JNK-1 on IL-4-induced gene expression. Macrophages were pre-incubated for 1 h with the JNK inhibitor SP600125 (5 μM) or vehicle (DMSO) as a control. The cells were then stimulated for 6 h with IL-4 except when gene expression of *SOCS1* (3 h) and *c-myc* and *p21^Waf1^* (1 h) were analyzed by qPCR. The results are shown as the mean ± SD of 3 independent experiments. * *p* < 0.05, ** *p* < 0.01, and *** *p* < 0.001 in relation to the corresponding treatments after all the independent experiments had been compared. Data were analyzed using Student’s *t*-test.

**Figure 4 cells-12-01127-f004:**
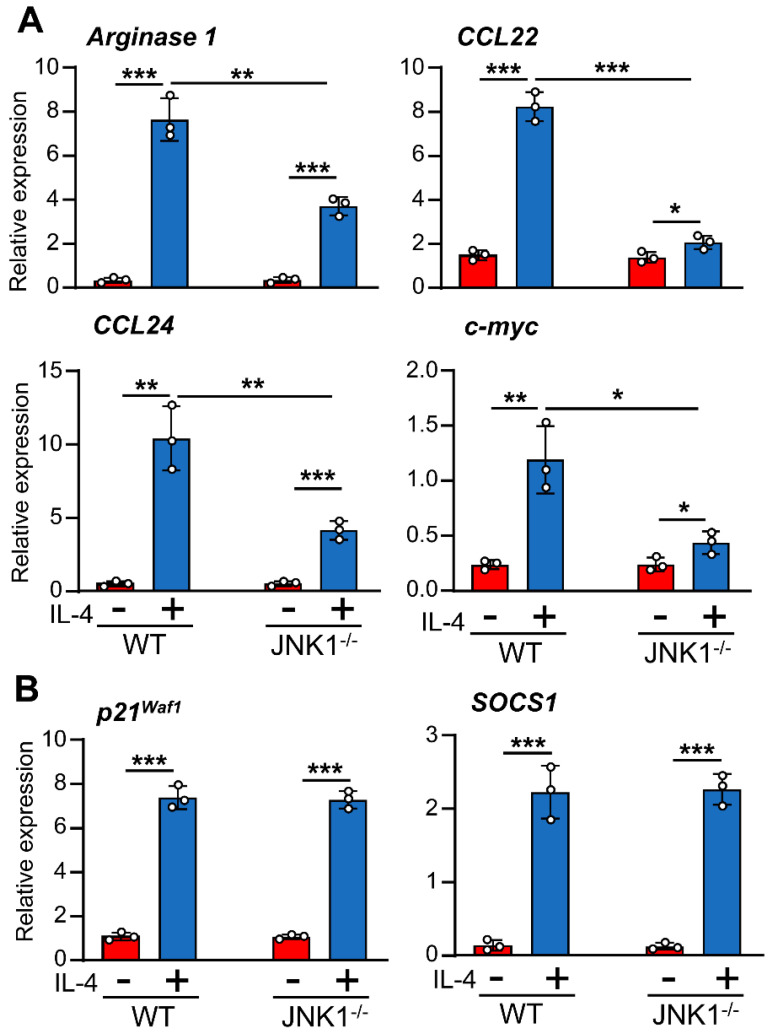
(**A**,**B**) Different effects of JNK-1 on IL-4-induced gene expression. Macrophages derived from WT or *JNK-1*-deficient mice (*JNK-1*^−/−^) were stimulated with IL-4 for 6h except when the gene expression of *SOCS1* (3 h), *p21^Waf−1^* (1 h), and c*-Myc* (1 h) was analyzed by qPCR. Control cells from each genotype were left untreated. The results are shown as the mean ± SD of 3 independent experiments. **p* < 0.05, ** *p* < 0.01, and *** *p* < 0.001 in relation to the corresponding treatments after all the independent experiments had been compared. Data were analyzed using Student’s *t*-test.

**Figure 5 cells-12-01127-f005:**
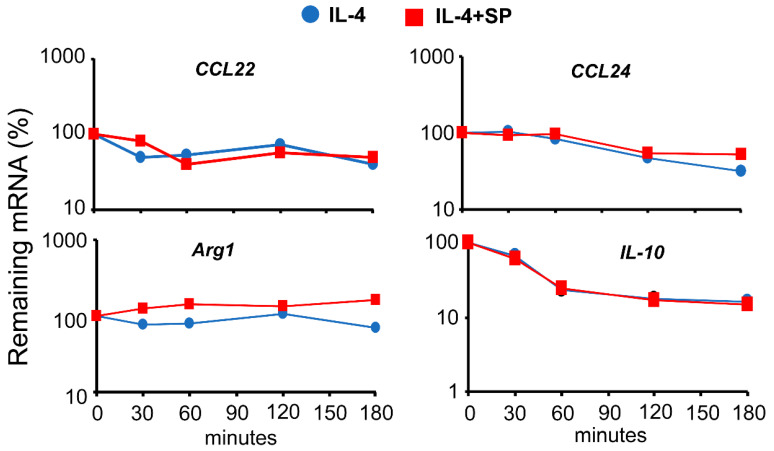
Effects of JNK-1 on the mRNA stability of IL-4-induced genes. Macrophages were pre-incubated with SP600125 for 1 h; then, IL-4 was added, and incubation proceeded for 6 h. At this point, a combination of the RNA synthesis inhibitors 5,6-dichlorobenzimidazole 1-β-D-ribofuranoside (DRB) (20 μg/mL) and actinomycin D (Act D) (5 μg/mL) was added for the indicated periods of time. The levels of gene expression were evaluated using qPCR. To evaluate the rate of mRNA degradation, the mRNA remaining after treatment with inhibitors of RNA synthesis was calculated as a percentage of the expression of the gene in the cells stimulated with IL-4 (+/− SP600125) in the absence of RNA synthesis inhibitors. These experiments were performed three times, and the results from the mean are shown. Data were analyzed using Student’s *t*-test, and no significant differences were found.

**Figure 7 cells-12-01127-f007:**
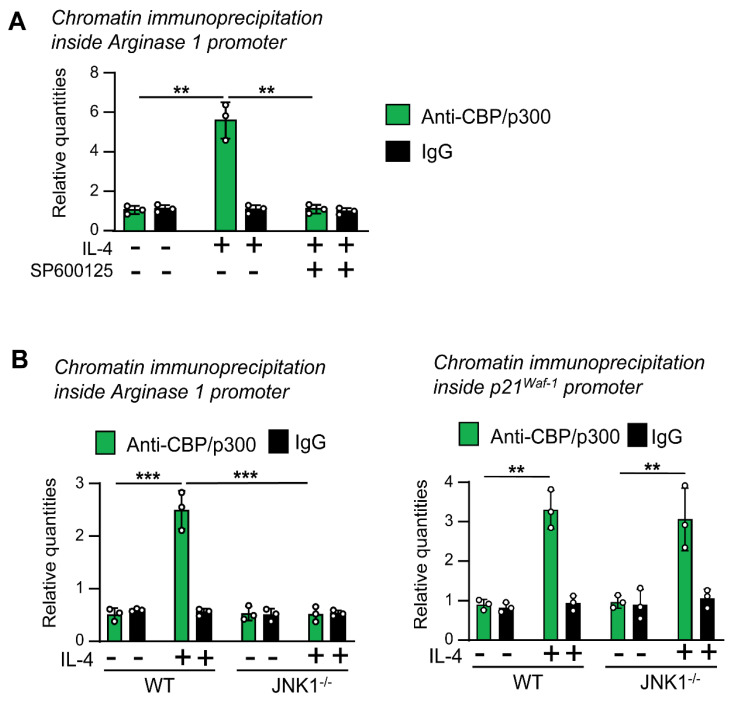
JNK-1 activity is required for the binding of the cofactor CBP/p300 to the *Arginase 1* promoter in response to IL-4 but not to the *p21^Waf−1^* promoter. Quiescent macrophages were treated with IL-4 for 15 min. In A, the cells were pretreated for 1 h with the JNK inhibitor SP600125 (SP) or the vehicle (DMSO) before the addition of IL-4. (**A**,**B**) Chromatin immunoprecipitation assay was performed with the antibodies indicated. The expression of the promoters was evaluated by quantitative PCR and normalized with the level of expression of a 36B4 exon and the inputs of each sample as a control for loading. The results are shown as the mean ± SD of 3 independent experiments. ** *p* < 0.01, and *** *p* < 0.001 in relation to the corresponding treatments after all the independent experiments had been compared. Data were analyzed using Student’s *t*-test.

## Data Availability

MDPI Research Data Policies.

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
