# Peer review of "Distinct Responses to IL4 in Macrophages Mediated by JNK"

_cells, 2023, doi:10.3390/cells12081127_

Round 1
Reviewer 1 Report
2 The manuscript of Arpa et al demonstrates that incubation of bone marrow-derived macrophages with IL-4 stimulates JNK-1. The phosphorylation of the other MAPKs ERK1, ERK2 and p38 occurs later or only weakly, respectively. The effect of IL-4 on MAPK phosphatases differs from M-CSF and induces the expression of MKP2 and MKP5. Pharmacological inhibition of JNK-1 with the inhibitor SP600125 reveals that expression of genes such as Arginase 1, CCL22 and others are impaired significantly while the expression p21Waf1 and SOCS1 is not influenced. JNK-1-deficient bone marrow-derived macrophages confirm these findings. The influence of JNK-1 on IL-4-induced gene expression is not exerted via a change in mRNA stability but relies on a change in transcription. Further, authors show that IL-4 induced phosphorylation of STAT-6 at Y641 could not be influenced by the JNk-1 inhibitor SP600125. Using chromatin immunoprecipitation the binding of STAT-6 to the Arginase 1 promoter was also independent of SP600125. However, the inhibitor influenced the IL-4 induced phosphorylation of STAT-6, and JNK-1-deficient bone marrow-derived macrophages confirm this finding. Finally, the binding of the cofactor CBP/p300 to the Arginase 1 promoter was prevented by the JNK-1 inhibitor SP600125 and this finding was again confirmed using JNK-1-deficient bone marrow-derived macrophages.
This is an interesting manuscript describing new mechanistic insight in the IL-4 induced M2 polarization of bone marrow-derived macrophages.
A few issues remain:
1) Line 150: “… was added to the three solutions.” It is unclear in my eyes to which three solutions the protease inhibitor cocktail was added.
2) Line 180/181: “… a “reverse crosslinking” step was required”. How was this done? By an incubation overnight at 65°C?
3) Figure 1 and line 211: I think the phosphorylation of ERK1 and ERK2 is quite substantial and not weak as described in line 211. This should be changed.
4) Lines 421 and 422: I am not sure what authors mean to say with this sentence. In my eyes, authors demonstrate that IL-4 triggers JNK-1 rapidly and other MAP kinases later or weakly. This is in my eyes a common signaling mechanism. If the following sentence in line 423 and 424 describes the uncommon signaling mechanism then the sentence in lines 423 and 424 should start with “Thus, we reported previously that …”.
5) Lines 433 to 435: Isn’t this sentence a contradiction to the previous sentence in lines 429 to 432? Authors state that crystal structures of STAT-6 bound to DNA show that S407 is not likely to be accessible for phosphorylation. This means in my eyes that the hypothesis of a conformational change of STAT-6 that makes S407 available for phosphorylation is very unlikely. I think the discussion of this issue should be improved.
6) Lines 454 and 455: This sentence appears to be incomplete and should be rephrased.
There are a number of minor issues, which should be addressed.
1) Line 50: “… family members of the were activated…” please replace by “… family members were activated…”.
2) Line 112: “… was accomplish…” please replace by “… was accomplished…”.
3) Line 140: “… 20 x 106 BMDMs…” please replace by “… 20 x 106(superscript) BMDMs …”.
4) Line 274: “… and c-Myc (1h) and gene…” please replace by “… and c-Myc (1h) gene …”.
5) Line 279: “JNK-1 do not affects the mRNA stability” replace by “JNK-1 does not affect the mRNA stability”
Author Response
Reviewer 1
The manuscript of Arpa et al demonstrates that incubation of bone marrow-derived macrophages with IL-4 stimulates JNK-1. The phosphorylation of the other MAPKs ERK1, ERK2 and p38 occurs later or only weakly, respectively. The effect of IL-4 on MAPK phosphatases differs from M-CSF and induces the expression of MKP2 and MKP5. Pharmacological inhibition of JNK-1 with the inhibitor SP600125 reveals that expression of genes such as Arginase 1, CCL22 and others are impaired significantly while the expression p21Waf1 and SOCS1 is not influenced. JNK-1-deficient bone marrow-derived macrophages confirm these findings. The influence of JNK-1 on IL-4-induced gene expression is not exerted via a change in mRNA stability but relies on a change in transcription. Further, authors show that IL-4 induced phosphorylation of STAT-6 at Y641 could not be influenced by the JNk-1 inhibitor SP600125. Using chromatin immunoprecipitation the binding of STAT-6 to the Arginase 1 promoter was also independent of SP600125. However, the inhibitor influenced the IL-4 induced phosphorylation of STAT-6, and JNK-1-deficient bone marrow-derived macrophages confirm this finding. Finally, the binding of the cofactor CBP/p300 to the Arginase 1 promoter was prevented by the JNK-1 inhibitor SP600125 and this finding was again confirmed using JNK-1-deficient bone marrow-derived macrophages.
This is an interesting manuscript describing new mechanistic insight in the IL-4 induced M2 polarization of bone marrow-derived macrophages.
A few issues remain:
Question 1. Line 150: “… was added to the three solutions.” It is unclear in my eyes to which three solutions the protease inhibitor cocktail was added.
Response: We modified the sentence as follows: “Before the centrifugation, a protease inhibitor cocktail (1 mM PMSF, 1 mM iodoacetam-ide, 1 mM sodium orthovanadate, 10 µg/mL of aprotinin, and 1 µg/mL of leupeptin) was added to the three solutions (0.1 M Tris-HCl, pH 9.4, containing 10 mM DTT; Buffer I and Buffer II).”
Question 2. Line 180/181: “… a “reverse crosslinking” step was required”. How was this done? By an incubation overnight at 65°C?
Response: The referee is correct and in order to clarified this question we modified the sentence as follows in the text: “Before DNA purification, a “reverse crosslinking” step was required by incubating the 300-µL samples (non-specific and immunoprecipitates) and INPUTs overnight at 65ºC.”
Question 3. Figure 1 and line 211: I think the phosphorylation of ERK1 and ERK2 is quite substantial and not weak as described in line 211. This should be changed.
Response: The sentence was modified as follows: In contrast to JNK-1, Western blot analysis of both phospho-ERK-1/2 and phospho-p38 revealed at earl times no activation but significant induction of both kinases after 60 minutes of IL-4 stimulation). (Figure 1B and C).
Question 4. Lines 421 and 422: I am not sure what authors mean to say with this sentence. In my eyes, authors demonstrate that IL-4 triggers JNK-1 rapidly and other MAP kinases later or weakly. This is in my eyes a common signaling mechanism. If the following sentence in line 423 and 424 describes the uncommon signaling mechanism then the sentence in lines 423 and 424 should start with “Thus, we reported previously that …”.
Response: We modified the sentence as follows: “Finally, we have been able to confirm our previous observations showing that gene induction by IL-4 does not have a common signaling mechanism Thus, we reported previously that deacetylation of C/EBPβ inhibited the IL-4 induced expression of Ar-ginase-1, Fizz1, and Mannose receptor, while in other genes, such as Ym1, Mgl1, and Mgl2, the expression was not affected [47].”
Question 5. Lines 433 to 435: Isn’t this sentence a contradiction to the previous sentence in lines 429 to 432? Authors state that crystal structures of STAT-6 bound to DNA show that S407 is not likely to be accessible for phosphorylation. This means in my eyes that the hypothesis of a conformational change of STAT-6 that makes S407 available for phosphorylation is very unlikely. I think the discussion of this issue should be improved.
Response: We modified the text as follows: “Recently, using multiple human cells lines of fibroblasts, activation of STAT6 has been shown to be critical in antiviral innate immunity [37]. In this case, the phosphorylation of serine 407 located in the DNA binding region plays a determining role. However, studies of the structural basis for DNA recognition by STAT6, show that the residue S407 is not likely to be accessible for phosphorylation by any kinase in the conformations of the protein observed in the crystal structures where STAT-6 is bound to DNA [48]. This controversy increases when the same authors showed that in luciferase reporter-based assays, the S407 mutation abolish IL-4 responses [48]. In fact, a large number of proteins including CBP/P300, CD28, C/EBPβ, Detergent-sensitive factor, Ets-1, glucocorticoid receptor (GR), IFNαRI, IL-4Rα, IRF4, LITAF, NF-kB, p100, PU.1, SRC-1 and STAT-2 has been reported to interact with STAT-6 [49] and the binding of some of these proteins may undergoes a conformational changes of STAT-6 that makes S407 available to be phosphorylated.”
Question 6. Lines 454 and 455: This sentence appears to be incomplete and should be rephrased.
Response: The referee is correct. We modified the sentence as follows: “The member of the p160/SRC family, SRC-1 (NCoA-1), was found to be crucial for ac-tivation by STAT-6.”
There are a number of minor issues, which should be addressed. Thanks to the referee for improving the manuscript.
1) Line 50: “… family members of the were activated…” please replace by “… family members were activated…”.
2) Line 112: “… was accomplish…” please replace by “… was accomplished…”.
3) Line 140: “… 20 x 106 BMDMs…” please replace by “… 20 x 106(superscript) BMDMs …”.
4) Line 274: “… and c-Myc (1h) and gene…” please replace by “… and c-Myc (1h) gene …”.f
5) Line 279: “JNK-1 do not affects the mRNA stability” replace by “JNK-1 does not affect the mRNA stability”
Reviewer 2 Report
Comments to the authors:
It is an ambitious attempt to understand the role IL-4 mediated signaling in M2 polarization with special emphasis on JNK1-mediated effects. I have given my comments below in no order.
1. There is no clarity on which STAT 6 serine residue is phosphorylated by JNK1, and it is important to discuss more because there are reports showing that Ser 707 phosphorylation of STAT 6 inactivates and decreases its binding.
2. Did you use M0 macrophages (unstimulated) as a control in your experiments in Fig?1?
3. Fig.1A, JNK1 in vitro activity assay using the immunoprecipitated JNK and c-Jun as substrate, is it the phosphorylation of c-Jun levels or just the expression of protein? Please give proper details in the text.
4. Fig.1B, at the 0-time point the Erk1 phosphorylation is higher than the 5 and 15 mins, is this decrease dependent on IL-4 or it is a late response as mentioned in the text?
5. Bar graphs aren't exactly matching with Fig1B and 1C. It is mentioned as a relative expression and with what? 0 point bar is random in all three images (Fig.1A, 1B, 1C).
6. Fig.3 and 4, JNK1 inhibition did not affect the phenotype of M2 macrophages as all the marker genes are significantly increased in presence of inhibitor vs control. Even when we compare DMSO +IL-4 and Inh + IL-4 the levels may be decreased but in the latter group with the control, significant change is there suggesting even JNK1 inhibited M2 phenotype did not affect. Please comment on this.
7. Fig.6A, IP for STAT6, as you mentioned is there no difference between the phosphorylation status of STAT6 Y641? 8. Fig.6C, how did you determine the phosphorylated serine is S407, text shows you used pSer antibody.
Author Response
Reviewer 2
It is an ambitious attempt to understand the role IL-4 mediated signaling in M2 polarization with special emphasis on JNK1-mediated effects. I have given my comments below in no order.
Question 1. There is no clarity on which STAT 6 serine residue is phosphorylated by JNK1, and it is important to discuss more because there are reports showing that Ser 707 phosphorylation of STAT 6 inactivates and decreases its binding.
Response: We modified the text as follows: “In the literature, IL-4 induce serine phosphorylation of STAT-6 is a highly controversial topic that depends a lot on the experimental conditions and, in particular, on the cell type used. Pesu et al. [48] using Ramos cells (a B cell of Burkitt's lymphoma origin) showed that IL-4 induce transcription require serine phosphorylation of Stat6. Shirakawa et al. [49] demonstrated in HeLa cells (a human cell from adenocarcinoma) that the cytokine IL-1 mediated by JNK induces STAT6 phosphorylation at serine 707. This phosphorylation decreases the DNA binding ability of IL-4-stimulated STAT6, been a mechanism of controlling the balance between IL-1 and IL-4 signals.”
Question 2. Did you use M0 macrophages (unstimulated) as a control in your experiments in Fig? 1?
Response: The referee is correct, time 0 means unstimulated
Question 3. Fig.1A, JNK1 in vitro activity assay using the immunoprecipitated JNK and c-Jun as substrate, is it the phosphorylation of c-Jun levels or just the expression of protein? Please give proper details in the text.
Response: We added the following in Material and Methods under 2.5. JNK activity assay:
“the reaction was performed with 1µg of cytosolic glutathione S-transferases (GST)-c-jun (1-169) (MBL) as JNK substrate”.
Also, under 3.1. IL-4 induces early and short activation of JNK-1 but not of ERK or p38, we added the following: “The activity of JNK-1, reported as glutathione S-transferases (GST)-c-jun, was strongly induced after 5 min of IL-4 treatment and was maintained for only 15 min (Figure 1A)”.
Question 4. Fig.1B, at the 0-time point the Erk1 phosphorylation is higher than the 5 and 15 mins, is this decrease dependent on IL-4 or it is a late response as mentioned in the text?
Response: These decreased values are due to individual variations of the method. When we analyzed the 3 independent experiments (Fig. 1B at the right), there are no significant variations.
Question 5. Bar graphs aren't exactly matching with Fig1B and 1C. It is mentioned as a relative expression and with what? 0 point bar is random in all three images (Fig.1A, 1B, 1C).
Response: Thanks to the referee. We added in the Figure 1A, “GST-c-Jun/JNK1”, in Figure 1B “ERK-1P/β-actin and ERK-2P/β-actin” and in Figure 1 C “p38P/β-actin”. O point is the control.
Question 6. Fig.3 and 4, JNK1 inhibition did not affect the phenotype of M2 macrophages as all the marker genes are significantly increased in presence of inhibitor vs control. Even when we compare DMSO +IL-4 and Inh + IL-4 the levels may be decreased but in the latter group with the control, significant change is there suggesting even JNK1 inhibited M2 phenotype did not affect. Please comment on this.
Response: In the following paragraph under 3.2. IL-4-induced JNK-1 activation contributes to the regulation of selective genes we explain that IL-4 responses are promoter-dependent:
“To determine the role of JNK-1 in IL-4-induced gene expression, we used the selec-tive inhibitor SP600125 as well as the JNK-1 knockout mice model. Previous studies in our group demonstrated that the dose of SP600125 used here in macrophages blocks JNK activity without inducing cellular toxicity [31]. Surprisingly, the inhibition of JNK with SP600125 resulted in efficient blockage of the expression of a subset of genes including Arginase 1, Mannose Receptor, CD163, and c-myc, the chemokines CCL22 and CCL24, and the cytokine IL-10 (Figure 3A), whereas the expression of SOCS1 or p21Waf-1 was not significantly reduced (Figure 3B), thereby suggesting that the link between JNK-1 and the IL-4 responses may be promoter-dependent.”
Also under Discussion there are the following comments: “Finally, we have been able to confirm our previous observations showing that gene induction by IL-4 does not have a common signaling mechanism Thus, we reported previously that deacetylation of C/EBPβ inhibited the IL-4 induced expression of Ar-ginase-1, Fizz1, and Mannose receptor, while in other genes, such as Ym1, Mgl1, and Mgl2, the expression was not affected [47].”
Question 7. Fig.6A, IP for STAT6, as you mentioned is there no difference between the phosphorylation status of STAT6 Y641?
Response: The referee is correct. In Fig. 6A and 6B we showed that inhibition of JNK do not affect phosphorylation status of STAT6 Y641. Therefore, JNK may affect other phosphorylation. In Fig. 6C we showed that serine phosphorylation was affected by JNK.
Question 8. Fig.6C, how did you determine the phosphorylated serine is S407, text shows you used pSer antibody.
Response: We use an antibody against serine phosphorylation. We do not have a specific antibody against S407. We end the paragraph with the following comment: “So far, all these data suggest that although JNK-1 mediates serine phosphorylation of STAT-6, it does not modify the phosphorylation of STAT-6 on tyrosine or its capacity to bind DNA.“ Therefore we do not know which serine is phosphorylated. In the Discussion we added the following: “One question that remains to be resolved is the location of the phosphorylated serine in STAT-6. Recently, activation of STAT6 has been shown to be critical in anti-viral innate immunity [37]. In this case, the phosphorylation of serine 407 located in the DNA binding region plays a determining role. However, studies of the structural basis for DNA recognition by STAT6, show that the residue S407 is not likely to be ac-cessible for phosphorylation by any kinase in the conformations of the protein ob-served in the crystal structures where STAT-6 is bound to DNA [48]. This controversy increases when in a luciferase reporter-based assay, the S407 mutation abolish IL-4 response. It is possible that the DNA-binding domain of STAT6 protein undergoes a conformational change that makes S407 available to be phosphorylated.”
Round 2
Reviewer 1 Report
Line 62 of the revised manuscript describes induction of ERKs as weak and late. I think they are induced late but significantly. This should be changed.
The paper still contains a number of grammar errors.
Reviewer 2 Report
NA